# One-2-3-45: Any Single Image to 3D Mesh in 45 Seconds without Per-Shape Optimization

**Minghua Liu**[1*]   **Chao Xu**[2*]   **Haian Jin**[3,4*]   **Linghao Chen**[1,4*]
**Mukund Varma T**[1]   **Zexiang Xu**[6]
**Hao Su**[1]

[1] UC San Diego  [2] UCLA  [3] Cornell University  [4] Zhejiang University  [5] Adobe Research

Project Website: `http://one-2-3-45.com`

## Abstract

Single image 3D reconstruction is an important but challenging task that requires extensive knowledge of our natural world. Many existing methods solve this problem by optimizing a neural radiance field under the guidance of 2D diffusion models but suffer from lengthy optimization time, 3D inconsistency results, and poor geometry. In this work, we propose a novel method that takes a single image of any object as input and generates a full 360-degree 3D textured mesh in a single feed-forward pass. Given a single image, we first use a view-conditioned 2D diffusion model, Zero123, to generate multi-view images for the input view, and then aim to lift them up to 3D space. Since traditional reconstruction methods struggle with inconsistent multi-view predictions, we build our 3D reconstruction module upon an SDF-based generalizable neural surface reconstruction method and propose several critical training strategies to enable the reconstruction of 360-degree meshes. Without costly optimizations, our method reconstructs 3D shapes in significantly less time than existing methods. Moreover, our method favors better geometry, generates more 3D consistent results, and adheres more closely to the input image. We evaluate our approach on both synthetic data and in-the-wild images and demonstrate its superiority in terms of both mesh quality and runtime. In addition, our approach can seamlessly support the text-to-3D task by integrating with off-the-shelf text-to-image diffusion models.

## 1 Introduction

Single image 3D reconstruction, the task of reconstructing a 3D model of an object from a single 2D image, is a long-standing problem in the computer vision community and is crucial for a wide range of applications, such as robotic object manipulation and navigation, 3D content creation, as well as AR/VR [47; 9; 92]. The problem is challenging as it requires not only the reconstruction of visible parts but also the hallucination of invisible regions. Consequently, this problem is often ill-posed and corresponds to multiple plausible solutions because of insufficient evidence from a single image. On the other hand, humans can adeptly infer unseen 3D content based on our extensive knowledge of the 3D world. To endow intelligent agents with this ability, many existing methods [31; 19; 25; 11; 87; 91; 16; 83; 39; 10; 37] exploit class-specific priors by training 3D generative networks on 3D shape datasets [4]. However, these methods often fail to generalize to unseen categories, and their reconstruction quality is constrained by the limited size of public 3D datasets.

---

*Equal Contribution

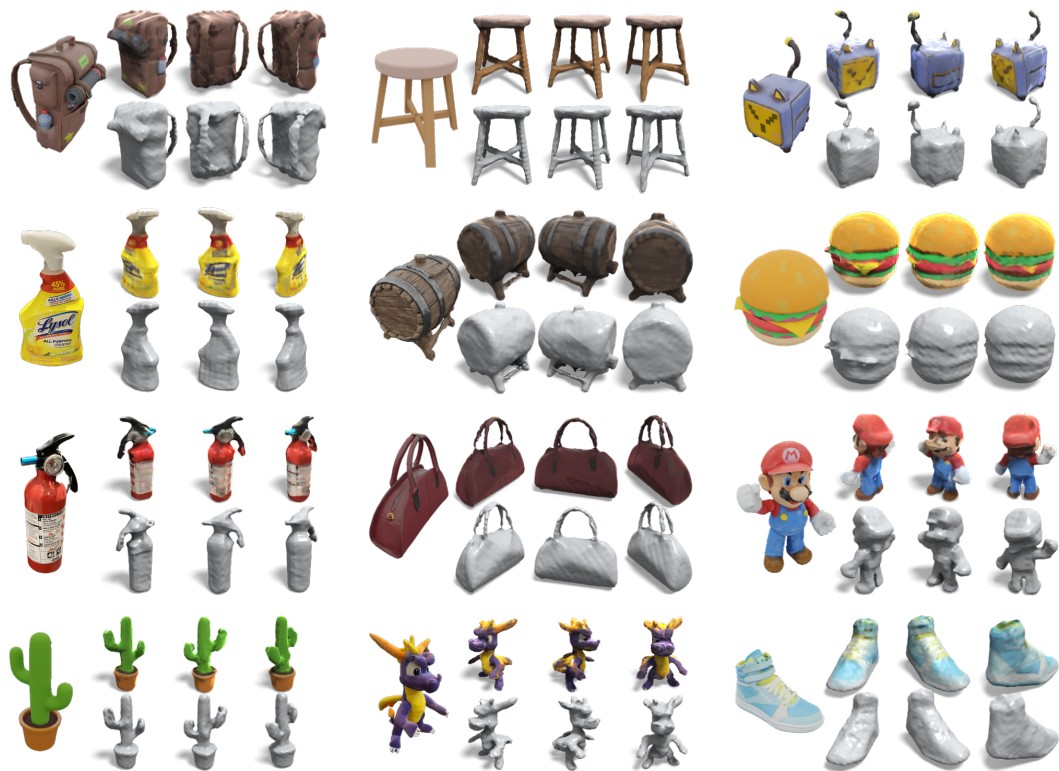

Figure 1: One-2-3-45 reconstructs a full 360° mesh of any object in 45 seconds given a single image of it. In each example, we showcase the input image in the left column, alongside the generated textured and textureless meshes from three different views.

In this work, we pursue a generic solution to turn an image of any object, regardless of its category, into a high-quality 3D textured mesh. To achieve this, we propose a novel approach that can effectively utilize the strong priors learned by 2D diffusion models for 3D reconstruction. Compared to 3D data, 2D images are more readily available and scalable. Recent 2D generative models (*e.g.*, DALL-E [64; 63], Imagen [69], and Stable Diffusion [68]) and visual-language models (*e.g.*, CLIP [61]) have made significant strides by pre-training on Internet-scale image datasets. Since they learn a wide range of visual concepts and possess strong priors about our 3D world, it is natural to marry 3D tasks with them. Consequently, an emerging body of research [26; 23; 52; 60; 36], as exemplified by DreamField [26], DreamFusion [60], and Magic3D [36], employs 2D diffusion models or vision language models to assist 3D generative tasks. The common paradigm of them is to perform per-shape optimization with differentiable rendering and the guidance of the CLIP model or 2D diffusion models. While many other 3D representations have been explored, neural fields are the most commonly used representation during optimization.

Although these optimization-based methods have achieved impressive results on both text-to-3D [60; 26; 36] and image-to-3D tasks [48; 72], they face some common dilemmas: (a) **time-consuming**. Per-shape optimization typically involves tens of thousands of iterations of full-image volume rendering and prior model inferences, resulting in typically tens of minutes per shape. (b) **memory intensive**. Since the full image is required for the 2D prior model, the volume rendering can be memory-intensive when the image resolution goes up. (c) **3D inconsistent**. Since the 2D prior model only sees a single view at each iteration and tries to make every view look like the input, they often generate 3D inconsistent shapes (*e.g.*, with two faces, or the Janus problem [48; 60]). (d) **poor geometry**. Many methods utilize the density field as the representation in volume rendering. It is common that they produce good RGB renderings but extracting high-quality mesh tends to be difficult.

In this paper, instead of following the common optimization-based paradigm, we propose a novel approach to utilize 2D prior models for 3D modeling. At the heart of our approach is the combination of a 2D diffusion model with a cost-volume-based 3D reconstruction technique, enabling

the reconstruction of a high-quality 360° textured mesh from a single image in a feed-forward pass without per-scene optimization. Specifically, we leverage a recent 2D diffusion model, Zero123 [41], which is fine-tuned on Stable Diffusion [68] to predict novel views of the input image given the camera transformation. We utilize it to generate multi-view predictions of the input single image so that we can leverage multi-view 3D reconstruction techniques to obtain a 3D mesh. There are two challenges associated with reconstruction from synthesized multi-view predictions: (a) the inherent lack of perfect consistency within the multi-view predictions, which can lead to severe failures in optimization-based methods such as NeRF methods [53; 5]. (b) the camera pose of the input image is required but unknown. To tackle them, we build our reconstruction module upon a cost volume-based neural surface reconstruction approach, SparseNeuS [45], which is a variant of MVSNeRF [6]. Additionally, we introduce a series of essential training strategies that enable the reconstruction of 360-degree meshes from inherently inconsistent multi-view predictions. We also propose an elevation estimation module that estimates the elevation of the input shape in Zero123's canonical coordinate system, which is used to compute the camera poses required by the reconstruction module.

By integrating the three modules of multi-view synthesis, elevation estimation, and 3D reconstruction, our method can reconstruct 3D meshes of any object from a single image in a feed-forward manner. Without costly optimizations, our method reconstructs 3D shapes in significantly less time, *e.g.*, in just 45 seconds. Our method favors better geometry due to the use of SDF representations, and generates more consistent 3D meshes, thanks to the camera-conditioned multi-view predictions. Moreover, our reconstruction adheres more closely to the input image compared to existing methods. See Figure 1 for some of our example results. We evaluate our method on both synthetic data and real images and demonstrate that our method outperforms existing methods in terms of both quality and efficiency.

## 2 Related Work

### 2.1 3D Generation Guided by 2D Prior Models

Recently, 2D generative models (*e.g.*, DALL-E [64; 63], Imagen [69], and Stable Diffusion [68]) and vision-language models (*e.g.*, CLIP [61]) have learned a wide range of visual concepts by pre-training on Internet-scale image datasets. They possess powerful priors about our 3D world and have inspired a growing body of research to employ 2D prior models for assisting 3D understanding [38; 40] and generative tasks. Exemplified by DreamField [26], DreamFusion [60], and Magic3D [36], a line of works follows the paradigm of per-shape optimization. They typically optimize a 3D representation (*i.e.*, NeRF, mesh, SMPL human model) and utilize differentiable rendering to generate 2D images from various views. The images are then fed to the CLIP model [23; 26; 52; 35; 3; 32; 2; 28; 89; 43] or 2D diffusion model [60; 36; 72; 48; 13; 78; 88; 51; 99; 62; 75] for calculating the loss functions, which are used to guide the 3D shape optimization. In addition to optimization-based 3D shape generation, some works train a 3D generative model but leverage the embedding space of CLIP [8; 44; 71], and some works focus on generating textures or materials for input meshes using 2D models' prior [52; 82; 7; 51; 67].

### 2.2 Single Image to 3D

Before the emergence of CLIP and large-scale 2D diffusion models, people often learn 3D priors from 3D synthetic data [4] or real scans [65]. Unlike 2D images, 3D data can be represented in various formats and numerous representation-specific 3D generative models have been proposed. By combining 2D image encoder and 3D generators, they generate 3D data in various representations, including 3D voxels [19; 85; 11; 87; 86; 91], point clouds [16; 94; 20; 1; 49; 96], polygon meshes [31; 79; 83; 56], and parametric models [59; 100; 101]. Recently, there has been an increasing number of work on learning to generate a 3D implicit field from a single image [90; 50; 70; 25; 58; 18; 21; 27; 55; 84; 54].

As previously mentioned, several recent works leverage 2D diffusion models to perform per-shape optimization, allowing for the text-to-3D task [60; 36; 26] given that diffusion models are typically conditioned on text. To enable the generation of 3D models from a single image, some works [48; 13; 51] utilize textual inversion [17], to find the best-matching text embedding for the input image, which is then fed into a diffusion model. NeuralLift-360 [24] adds a CLIP loss to enforce similarity between the rendered image and the input image. 3DFuse [72] finetunes the Stable Diffusion model with LoRA layers [24] and a sparse depth injector to ensure greater 3D consistency. A recent work

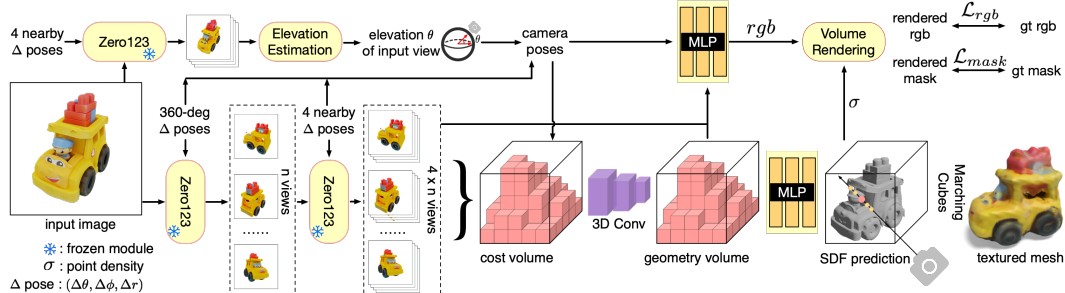

Figure 2: Our method consists of three primary components: (a) **Multi-view synthesis**: we use a view-conditioned 2D diffusion model, Zero123 [41], to generate multi-view images in a two-stage manner. The input of Zero123 includes a single image and a relative camera transformation, which is parameterized by the relative spherical coordinates $(\Delta\theta, \Delta\phi, \Delta r)$. (b) **Pose estimation**: we estimate the elevation angle $\theta$ of the input image based on four nearby views generated by Zero123. We then obtain the poses of the multi-view images by combining the specified relative poses with the estimated pose of the input view. (c) **3D reconstruction**: We feed the multi-view posed images to an SDF-based generalizable neural surface reconstruction module for $360°$ mesh reconstruction.

Zero123 [41; 73] finetunes the Stable Diffusion model [69] to generate a novel view of the input image based on relative camera pose. In addition to these methods, OpenAI trains a 3D native diffusion model Point-E [57], which uses several million internal 3D models to generate point clouds. Very recently, they published another model Shap-E [30] which is trained to generate parameters of implicit functions that can be used for producing textured meshes or neural radiance fields.

## 2.3 Generalizable Neural Reconstruction

Traditional NeRF-like methods [53; 80] use a neural network to represent a single scene and require per-scene optimization. However, some approaches aim to learn priors across scenes and generalize to novel scenes. These methods typically take a few source views as input and leverage 2D networks for extracting 2D features. The pixel features are then unprojected into 3D space, and a NeRF-based rendering pipeline is applied on top of them. In this way, they can generate a 3D implicit field given a few source views in a single feed-forward pass. Among the methods, some [81; 65; 22; 95; 93; 42; 34; 76; 77] directly aggregate 2D features with MLPs or transformers, while others explicitly construct the 3D feature/cost volume [6; 29; 98; 45], and utilize the voxel feature for decoding density and color. In addition to the density field representation, some methods such as SparseNeuS [45] and VolRecon [66] utilize SDF representations for geometry reconstruction.

## 3 Method

Our overall pipeline is illustrated in Figure 2. In Section 3.1, we introduce a view-conditioned 2D diffusion model, Zero123 [41], which is used to generate multi-view images. In Section 3.2, we show that traditional NeRF-based and SDF-based methods fail to reconstruct high-quality meshes from inconsistent multi-view predictions even given ground truth camera poses. Therefore, in Section 3.3, we propose a cost volume-based neural surface reconstruction module that can be trained to handle inconsistent multi-view predictions and reconstruct a 3D mesh in a single feed-forward pass. Specifically, we build upon the SparseNeuS [45] and introduce several critical training strategies to support $360°$ mesh reconstruction. Additionally, in Section 3.4, we demonstrate the necessity of estimating the pose of the input view in Zero123's canonical space for 3D reconstruction. While the azimuth and radius can be arbitrarily specified, we propose a novel module that utilizes four nearby views generated by Zero123 to estimate the elevation of the input view.

## 3.1 Zero123: View-Conditioned 2D Diffusion

Recent 2D diffusion models [64; 69; 68] have demonstrated the ability to learn a wide range of visual concepts and strong priors by training on internet-scale data. While the original diffusion

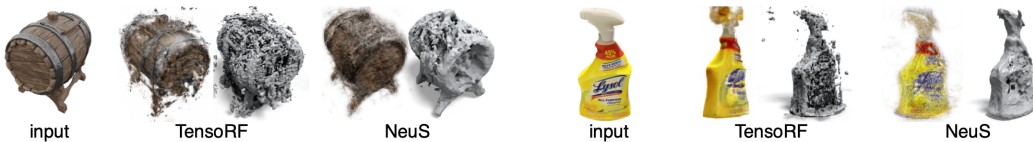

input      TensoRF      NeuS      input      TensoRF      NeuS

Figure 3: NeRF-based method [53] and SDF-based method [80] fail to reconstruct high-quality meshes given multi-view images predicted by Zero123. See Figure 1 for our reconstruction results.

models mainly focused on the task of text-to-image, recent work [97; 24] has shown that fine-tuning pretrained models allows us to add various conditional controls to the diffusion models and generate images based on specific conditions. Several conditions, such as canny edges, user scribbles, depth, and normal maps, have already proven effective [97].

The recent work Zero123 [41] shares a similar spirit and aims to add viewpoint condition control for the Stable Diffusion model [68]. Specifically, given a single RGB image of an object and a relative camera transformation, Zero123 aims to control the diffusion model to synthesize a new image under this transformed camera view. To achieve this, Zero123 fine-tunes the Stable Diffusion on paired images with their relative camera transformations, synthesized from a large-scale 3D dataset [12]. During the creation of the fine-tuning dataset, Zero123 assumes that the object is centered at the origin of the coordinate system and uses a spherical camera, *i.e.*, the camera is placed on the sphere's surface and always looks at the origin. For two camera poses $(\theta_1, \phi_1, r_1)$ and $(\theta_2, \phi_2, r_2)$, where $\theta_i$, $\phi_i$, and $r_i$ denote the polar angle, azimuth angle, and radius, their relative camera transformation is parameterized as $(\theta_2 - \theta_1, \phi_2 - \phi_1, r_2 - r_1)$. They aim to learn a model $f$, such that $f(x_1, \theta_2 - \theta_1, \phi_2 - \phi_1, r_2 - r_1)$ is perceptually similar to $x_2$, where $x_1$ and $x_2$ are two images of an object captured from different views. Zero123 finds that such fine-tuning enables the Stable Diffusion model to learn a generic mechanism for controlling the camera viewpoints, which extrapolates outside of the objects seen in the fine-tuning dataset.

## 3.2 Can NeRF Optimization Lift Multi-View Predictions to 3D?

Given a single image of an object, we can utilize Zero123 [41] to generate multi-view images, but can we use traditional NeRF-based or SDF-based methods [5; 80] to reconstruct high-quality 3D meshes from these predictions? We conduct a small experiment to test this hypothesis. Given a single image, we first generate 32 multi-view images using Zero123, with camera poses uniformly sampled from the sphere surface. We then feed the predictions to a NeRF-based method (TensoRF [53]) and an SDF-based method (NeuS [80]), which optimize density and SDF fields, respectively. However, as shown in Figure 3, both methods fail to produce satisfactory results, generating numerous distortions and floaters. This is primarily due to the inconsistency of Zero123's predictions. In Figure 4, we compare Zero123's predictions with ground-truth renderings. We can see that the overall PSNR is not very high, particularly when the input relative pose is large or the target pose is at unusual locations (*e.g.*, from the bottom or the top). However, the mask IoU (most regions are greater than 0.95) and CLIP similarity are relatively good. This suggests that Zero123 tends to generate predictions that are perceptually similar to the ground truth and have similar contours or boundaries, but the pixel-level appearance may not be exactly the same. Nevertheless, such inconsistencies between the source views are already fatal to traditional optimization-based methods. Although the original Zero123 paper proposes another method for lifting its multi-view predictions, we will demonstrate in experiments that it also fails to yield perfect results and entails time-consuming optimization.

## 3.3 Neural Surface Reconstruction from Imperfect Multi-View Predictions

Instead of using optimization-based approaches, we base our reconstruction module on a generalizable SDF reconstruction method SparseNeuS [45], which is essentially a variant of the MVSNeRF [6] pipeline that combines multi-view stereo, neural scene representation, and volume rendering. As illustrated in Figure 2, our reconstruction module takes multiple source images with corresponding camera poses as input and generates a textured mesh in a single feed-forward pass. In this section, we will first briefly describe the network pipeline of the module and then explain how we train the module, select the source images, and generate textured meshes. Additionally, in Section 3.4, we will discuss how we generate the camera poses for the source images.

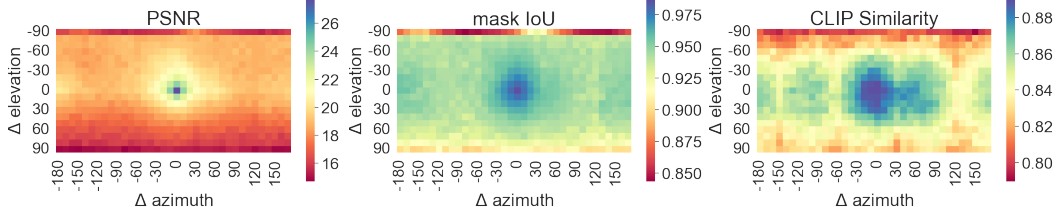

Figure 4: We analyze the prediction quality of Zero123 by comparing its predictions to ground truth renderings across various view transformations. For each view transformation, we report the average PSNR, mask IoU, and CLIP similarity of 100 shapes from the Objaverse [12] dataset. The prediction mask is calculated by considering foreground objects (*i.e.*, non-white regions). Zero123 provides more accurate predictions when the view transformation is small.

As shown in Figure 2, our reconstruction module takes $m$ posed source images as input. The module begins by extracting $m$ 2D feature maps using a 2D feature network. Next, the module builds a 3D cost volume whose contents are computed by first projecting each 3D voxel to $m$ 2D feature planes and then fetching the variance of the features across the $m$ projected 2D locations. The cost volume is then processed using a sparse 3D CNN to obtain a geometry volume that encodes the underlying geometry of the input shape. To predict the SDF at an arbitrary 3D point, an MLP network takes the 3D coordinate and its corresponding interpolated features from the geometry encoding volume as input. To predict the color of a 3D point, another MLP network takes as input the 2D features at the projected locations, interpolated features from the geometry volume, and the viewing direction of the query ray relative to the viewing direction of the source images. The network predicts the blending weights for each source view, and the color of the 3D point is predicted as the weighted sum of its projected colors. Finally, an SDF-based rendering technique is applied on top of the two MLP networks for RGB and mask rendering [80]. In each iteration, we randomly choose one view to build the cost volume and another view for rendering supervision.

**2-Stage Source View Selection and Groundtruth-Prediction Mixed Training.** Although the original SparseNeuS [45] paper only demonstrated frontal view reconstruction, we have extended it to reconstruct 360-degree meshes in a single feed-forward pass by selecting source views in a particular way. Specifically, our reconstruction model is trained on a 3D object dataset while freezing Zero123. We follow Zero123 to normalize the training shapes and use a spherical camera model. For each shape, we first render $n$ ground-truth RGB images from $n$ camera poses uniformly placed on the sphere. For each of the $n$ views, we use Zero123 to predict four nearby views. During training, we feed all $4 \times n$ predictions with ground-truth poses into the reconstruction module and randomly choose one of the $n$ ground-truth RGB images views as the target view. We call this view selection strategy as *2-stage source view selection*. We supervise the training with both the ground-truth RGB and mask values. In this way, the module can learn to handle the inconsistent predictions from Zero123 and reconstruct a consistent $360°$ mesh. We argue that our two-stage source view selection strategy is critical since uniformly choosing $n \times 4$ source views from the sphere surface would result in larger distances between the camera poses. However, cost volume-based methods [45; 29; 6] typically rely on very close source views to find local correspondences. Furthermore, as shown in Figure 4, when the relative pose is small (*e.g.*, 10 degrees apart), Zero123 can provide very accurate and consistent predictions and thus can be used to find local correspondences and infer the geometry.

During training, we utilize $n$ ground-truth renderings in the initial stage. We find that employing $n$ predicted images at this stage would suffer from notable inconsistencies across different views, complicating the network's ability to learn sharp details (see examples in ablation study). However, during inference, we can replace the $n$ ground-truth renderings with Zero123 predictions, as shown in Figure 2, the network can automatically generalize to some extent. We will show in the experiments that this groundtruth-prediction mixed training strategy is also important. To export the textured mesh, we use marching cubes [46] to extract the mesh from the predicted SDF field and query the color of the mesh vertices as described in [80]. Although our reconstruction module is trained on a 3D dataset, we find that it mainly relies on local correspondences and can generalize to unseen shapes very well.

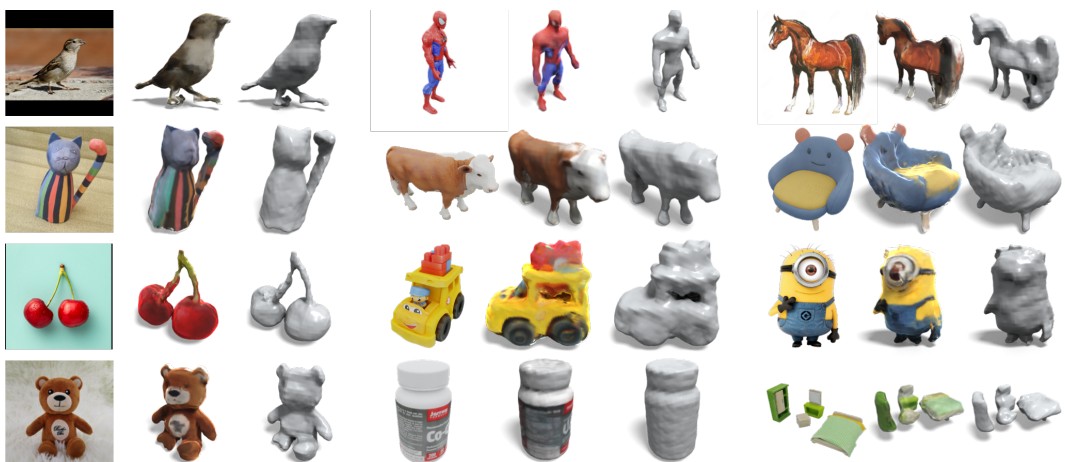

Figure 5: Qualitative examples of One-2-3-45 for both synthetic and real images. Each triplet showcases an input image, a textured mesh, and a textureless mesh.

## 3.4 Camera Pose Estimation

Our reconstruction module requires camera poses for the $4 \times n$ source view images. Note that we adopt Zero123 for image synthesis, which parameterizes cameras in a canonical spherical coordinate frame, $(\theta, \phi, r)$, where $\theta$, $\phi$ and $r$ represent the elevation, azimuth, and radius. While we can arbitrarily adjust the azimuth angle $\phi$ and the radius $r$ of all source view images simultaneously, resulting in the rotation and scaling of the reconstructed object accordingly, this parameterization requires knowing the absolute elevation angle $\theta$ of one camera to determine the relative poses of all cameras in a standard XYZ frame. More specifically, the relative poses between camera $(\theta_0, \phi_0, r_0)$ and camera $(\theta_0 + \Delta\theta, \phi_0 + \Delta\phi, r_0)$ vary for different $\theta_0$ even when $\Delta\theta$ and $\Delta\phi$ are the same. Because of this, changing the elevation angles of all source images together (*e.g.*, by 30 degrees up or 30 degrees down) will lead to the distortion of the reconstructed shape (see Figure 10 for examples).

Therefore, we propose an elevation estimation module to infer the elevation angle of the input image. First, we use Zero123 to predict four nearby views of the input image. Then we enumerate all possible elevation angles in a coarse-to-fine manner. For each elevation candidate angle, we compute the corresponding camera poses for the four images and calculate a reprojection error for this set of camera poses to measure the consistency between the images and the camera poses. The elevation angle with the smallest reprojection error is used to generate the camera poses for all $4 \times n$ source views by combining the pose of the input view and the relative poses. Please refer to the appendix for details on how we calculate the reprojection error for a set of posed images.

## 4 Experiments

### 4.1 Implementation Details

For each input image, we generate $n = 8$ images by choosing camera poses uniformly placed on the sphere surface and then generate 4 local images ($10°$ apart) for each of the 8 views, resulting in 32 source-view images for reconstruction. During training, we freeze the Zero123 [41] model and train our reconstruction module on the Objaverse-LVIS [12] dataset, which contains 46K 3D models in 1,156 categories. We use BlenderProc [14] to render ground-truth RGB images. For images with background, we utilize an off-the-shelf segmentation network SAM [33] with bounding-box prompts for background removal. Please refer to the appendix for more details.

### 4.2 Single Image to 3D Mesh

We present qualitative examples of our method in Figures 1 and 5, illustrating its effectiveness in handling both synthetic images and real images. We also compare One-2-3-45 with existing zero-shot single image 3D reconstruction approaches, including Point-E [57], Shap-E [30], Zero123 (Stable

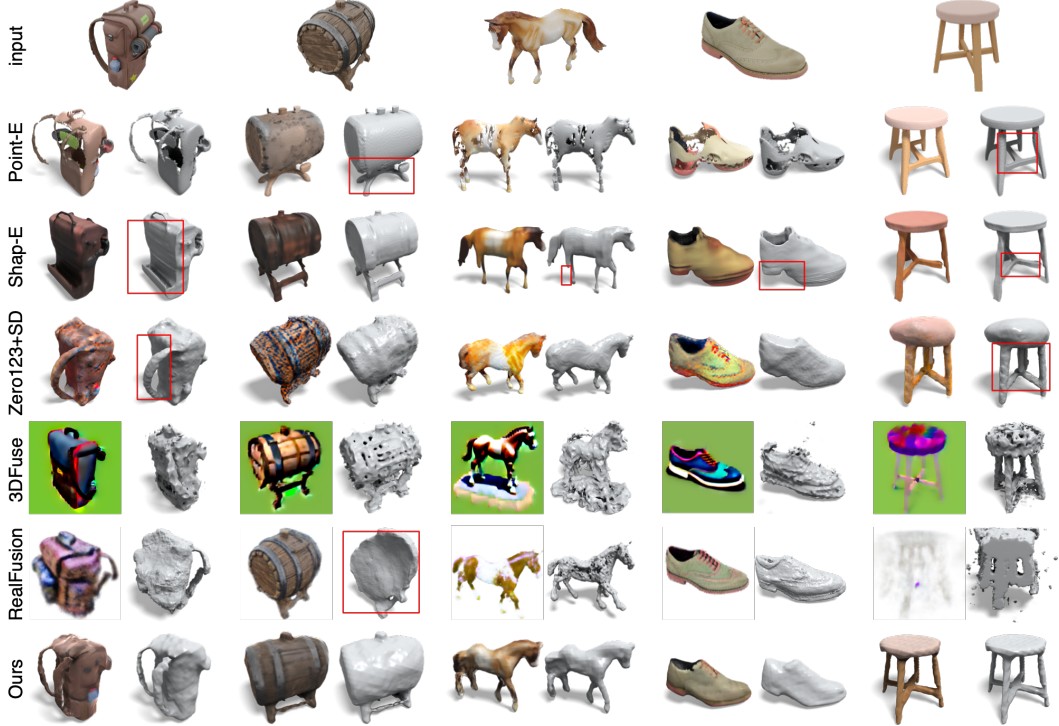

Figure 6: We compare One-2-3-45 with Point-E [57], Shap-E [30], Zero123 (Stable Dreamfusion version) [41], 3DFuse [72], and RealFusion [48]. In each example, we present both the textured and textureless meshes. As 3DFuse [72] and RealFusion [48] do not natively support the export of textured meshes, we showcase the results of volume rendering instead.

Table 1: Quantitative Comparison on GSO [15] and Objaverse [12] datasets.

| | Prior Source | F-Score | | | CLIP Similarity | | | Time |
|---|---|---|---|---|---|---|---|---|
| | | GSO | Obj. | avg. | GSO | Obj. | avg. | |
| Point-E [57] | internal | 81.0 | 81.0 | 81.0 | 74.3 | 78.5 | 76.4 | 78s |
| Shap-E [30] | 3D data | 83.4 | 81.2 | 82.3 | 79.6 | 82.1 | 80.9 | 27s |
| Zero123+SD [41] | 2D diffusion models | 75.1 | 69.9 | 72.5 | 71.0 | 72.7 | 71.9 | ~15min |
| RealFusion [48] | | 66.7 | 59.3 | 63.0 | 69.3 | 69.5 | 69.4 | ~90min |
| 3DFuse [72] | | 60.7 | 60.2 | 60.4 | 71.4 | 74.0 | 72.7 | ~30min |
| Ours | | 84.0 | 83.1 | 83.5 | 76.4 | 79.7 | 78.1 | 45s |

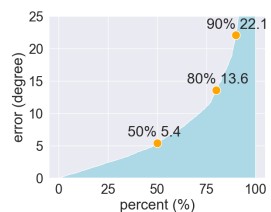

Figure 7: Error distribution of predicted elevations. The median and average are 5.4 and 9.7 degrees.

Dreamfusion version) [41], 3DFuse [72], and RealFusion [48]. Among them, Point-E and Shap-E are two 3D native diffusion models released by OpenAI, which are trained on several million internal 3D data, while others are optimization-based approaches leveraging priors from Stable Diffusion [68].

Figure 6 presents the qualitative comparison. While most methods can generate plausible 3D meshes from a single image, notable differences exist among them in terms of geometry quality, adherence to the input, and overall 3D consistency. In terms of geometry quality, approaches like RealFusion [48] and 3DFuse [72], which optimize a neural radiance field, face challenges in extracting high-quality meshes. Likewise, Point-E [57] produces a sparse point cloud as its output, resulting in numerous holes on the reconstructed meshes. In contrast, our approach utilizes an SDF presentation and favors better geometry. Regarding adherence to the input, we observe that most baseline methods struggle to preserve the similarity to the input image. Although Shap-E performs slightly better, it still produces lots of failure cases (see the backpack without shoulder straps, distorted shoe, and stool with three legs). In contrast, our approach leverages a powerful 2D diffusion model to directly produce high-quality multi-view images, rather than relying on 3D space hallucination. This strategy provides

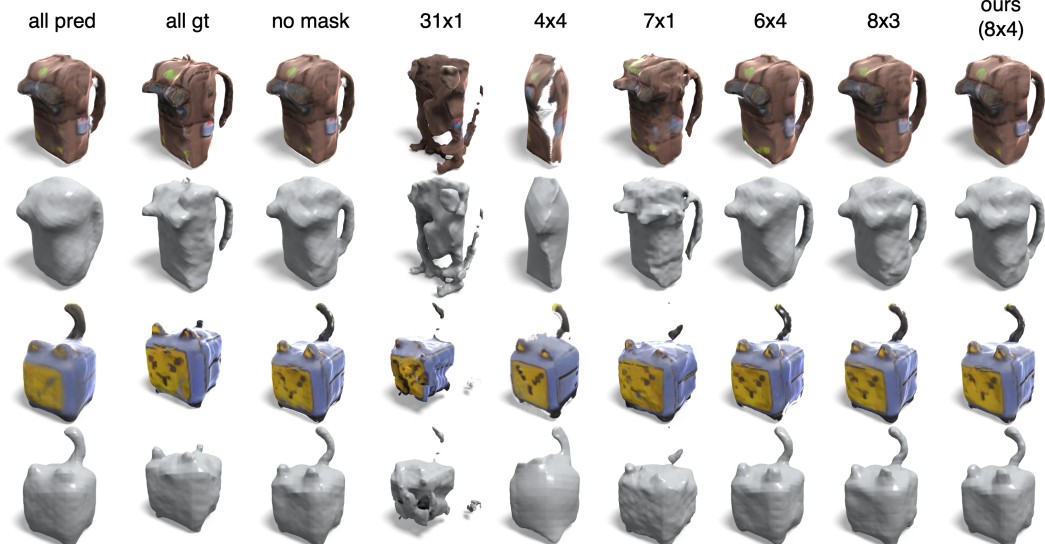

| all pred | all gt | no mask | 31x1 | 4x4 | 7x1 | 6x4 | 8x3 | ours (8x4) |

Figure 8: Ablations on training strategies of the reconstruction module and the number of views.

better adherence to the input views, alleviates the burden of the 3D reconstruction module, and yields results that are more finely attuned to the input. Furthermore, many approaches encounter challenges in achieving consistent 3D results (also known as the Janus problem [48; 60]), as highlighted in the right figure (two-handle mug, multi-face Mario, and two-face backpack). One of the contributing factors to this issue is that several methods optimize each view independently, striving to make each view resemble the input. In contrast, our method capitalizes on the view-conditioned 2D diffusion model, inherently enhancing 3D consistency.

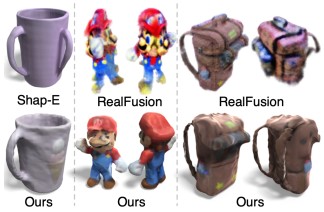

We also quantitatively compare the approaches on Objaverse [12] and GoogleScannedObjects (GSO) [15] datasets. For each dataset, we randomly choose 20 shapes and render a single image per shape for evaluation. To align the predictions with the ground-truth mesh, we linearly search the scaling factor and the rotation angle, apply Iterative Closest Point (ICP) for sampled point clouds, and select the one with the most number of inliers. We follow RealFusion [48] to report F-score (with a threshold of 0.05) and CLIP similarity, and the runtime on an A100 GPU. As shown in Table 1, our method outperforms all baseline approaches in terms of F-Score. As for CLIP similarity, we surpass all methods except a concurrent work Shap-E [30]. We find that CLIP similarity is very sensitive to the color distribution and less discriminative in local geometry variations (*i.e.*, the number of legs of a stool, the number of handles of a mug). Regarding running time, our method demonstrates a notable advantage over optimization-based approaches and performs on par with 3D native diffusion models, such as Point-E [57] and Shap-E [30]. Specifically, our 3D reconstruction module reconstructs a 3D mesh in approximately 5 seconds, with the remaining time primarily spent on Zero123 predictions, which takes roughly 1 second per image on an A100 GPU.

### 4.3 Ablation Study

**Training strategies.** We ablate our training strategies in Figure 8. We found that without our 2-stage source view selection strategy, a network trained to consume 31 uniformly posed Zero123 predictions (fourth column) suffers from severe inconsistency among source views, causing the reconstruction module to fail completely. If we feed only 7 source views (sixth column) without the four nearby views, the reconstruction fails to capture local correspondence and cannot reconstruct fine-grained geometry. During training, we first render $n$ ground-truth renderings and then use Zero123 to predict four nearby views for each of them. If we train directly on $8 \times 4$ ground-truth renderings without Zero123 prediction during training (second column), it fails to generalize well to Zero123 predictions

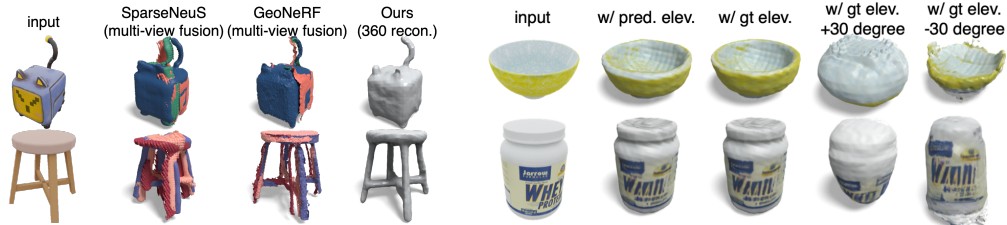

Figure 9: 360° reconstruction vs. multi-view fusion. Meshes from different views are in different colors.

Figure 10: Incorrect elevations lead to distorted reconstruction. Our elevation estimation module can predict an accurate elevation of the input view.

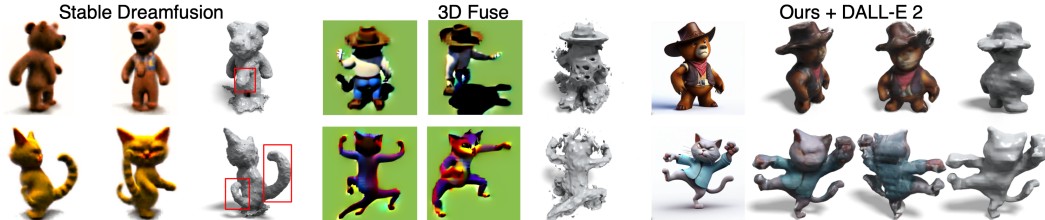

Figure 11: Text to 3D. First row: "a bear in cowboy suit." Second row: "a kungfu cat." We utilize DALL-E 2 [63] to generate an image conditioned on the text and then lift it to 3D. We compare our method with Stable Dreamfusion [60] and 3DFuse [72]. For baselines, volume renderings are shown.

during inference, with many missing regions. Instead, if we replace the $n$ ground-truth renderings with $n$ Zero123 predictions during training (first column), the network fail to generate sharp details (see the strips of the backpack).

**Elevation estimation.** Our reconstruction module relies on accurate elevation angles of the input view. In Figure 10, we demonstrate the impact of providing incorrect elevation angles (*e.g.*, altering the elevation angles of source views by $\pm 30°$), which results in distorted reconstruction results. Instead, utilizing our predicted elevation angles can perfectly match results with ground truth elevations. We also quantitatively test our elevation estimation module by rendering 1,700 images from random camera poses. As shown in Figure 7, our elevation estimation module predicts accurate elevations.

**Number of source views.** In Figure 8, we also investigate the impact of varying the number of source views on 3D reconstruction. We observe that our method is not very sensitive to the number of views as long as the reconstruction module is retrained with the corresponding setting.

360° **reconstruction vs. multi-view fusion.** While our method reconstructs a 360° mesh in a single pass, most existing generalizable neural reconstruction approaches [45; 29; 6] primarily focus on frontal view reconstruction. An alternative approach is to independently infer the geometry for each view and subsequently fuse them together. However, we have observed that this strategy often struggles with multi-view fusion due to inconsistent Zero123 predictions, as illustrated in Figure 9.

### 4.4 Text to 3D Mesh

As shown in Figure 11, by integrating with off-the-shelf text-to-image 2D diffusion models [68; 63], our method can be naturally extended to support text-to-image-3D tasks and generate high-quality textured meshes in a short time. See supplementary for more examples.

## 5 Conclusion

In this paper, we present a novel method for reconstructing a high-quality 360° mesh of any object from a single image of it. In comparison to existing zero-shot approaches, our results exhibit superior geometry, enhanced 3D consistency, and a remarkable adherence to the input image. Notably, our approach reconstructs meshes in a single forward pass without the need for time-consuming optimization, resulting in significantly reduced processing time. Furthermore, our method can be effortlessly extended to support the text-to-3D task.

## Acknowledgments

This work is supported in part by gifts from Qualcomm. We would like to thank Ruoxi Shi, Xinyue Wei, Hansheng Chen, Jiayuan Gu, Fanbo Xiang, Xiaoshuai Zhang, and Yulin Liu for their helpful discussions and manuscript proofreading.

We would like to thank the following sketchfab users for the models used for the demo images in this paper: dimaponomar2019 (backpack), danielpeng (bag), pmlzbt233 (wooden barrel), felixyadomi (cactus), avianinda (burger), shedmon (robocat), ie-niels (stool), phucn (armchair), techCIR (mug), sabriny (fox). All models are CC-By licensed.

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

## Appendix

We first show more qualitative comparison in Section A, which is followed by a demonstration of additional examples on real-world images and the text-to-3D task in Sections B and C respectively. Furthermore, we present the details of our elevation estimation module in Section D, training and evaluation details in Section E. We finally show the failure cases and discuss the limitations in Section F.

## A    More Qualitative Comparison

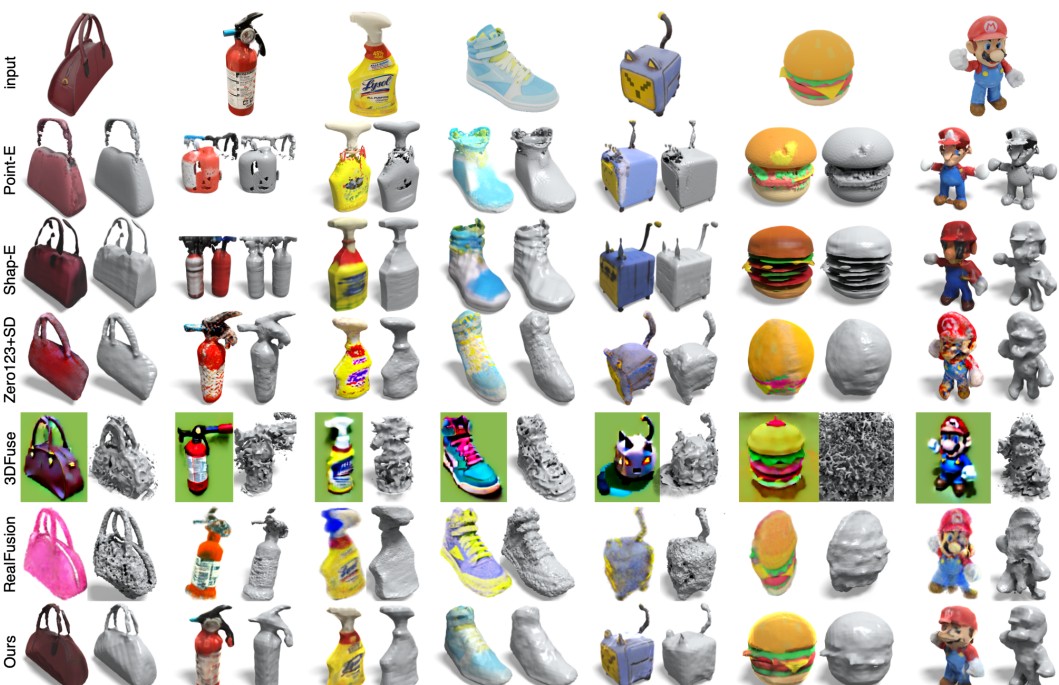

Figure 12: We compare One-2-3-45 with Point-E [57], Shap-E [30], Zero123 (Stable Dreamfusion version) [41], 3DFuse [72], and RealFusion [48]. In each example, we present both the textured and textureless meshes. As 3DFuse [72] and RealFusion [48] do not natively support the export of textured meshes, we showcase the results of volume rendering instead.

In Figure 12, we demonstrate more qualitative comparison on Objaverse [12] and GoogleScannedObjects (GSO) [15] datasets. Note that all test shapes are not seen during the training of our 3D reconstruction module.

## B    More Examples on Real-World Images

In Figure 13, we showcase more examples on real-world images and compare our method with the concurrent method Shap-E [30]. The input images are from `unsplash.com` or captured by ourselves. Note that our results exhibit a closer adherence to the input image.

## C    More Examples on Text-to-3D

In Figure 14, we present additional examples for the text-to-3D task. It is evident that existing approaches struggle to capture fine-grained details, such as a tree hollow, or achieve compositionality, as seen in examples like an orange stool with green legs, a pineapple-shaped Havana hat, or a rocking horse chair. In contrast, our method produces superior results that adhere more closely to the input text. We hypothesize that controlling such fine-grained attributes in the 3D space using existing optimization strategies is inherently challenging. However, by leveraging established 2D

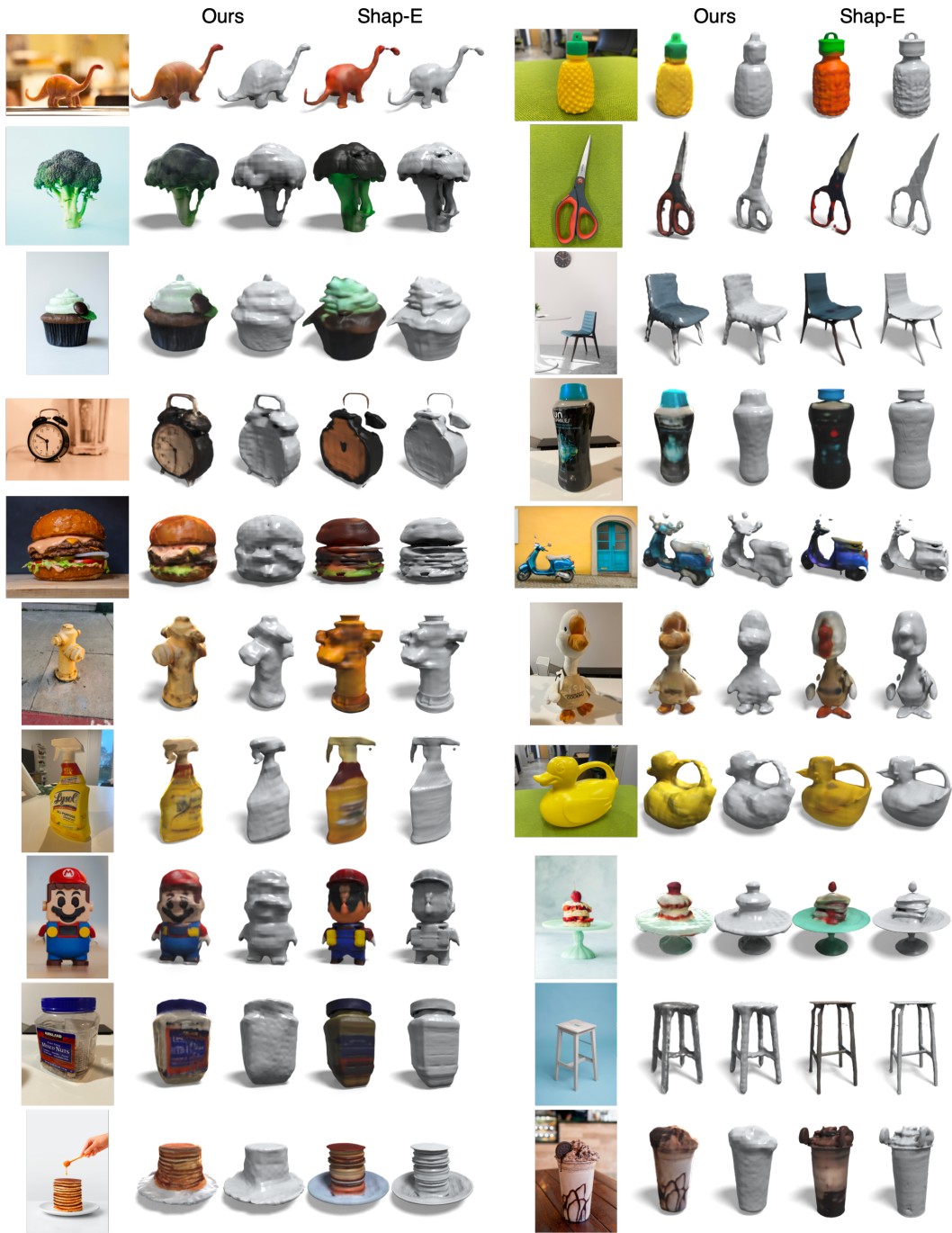

Figure 13: We compare One-2-3-45 with Shap-E [30] on real-world images. In each example, we present the input image, generated textured and textureless meshes.

text-to-image diffusion models, our method becomes more effective in lifting a single 2D image to a corresponding 3D textured mesh.

## D  Details of Elevation Estimation

To estimate the elevation angle $\theta$ of the input image, we first utilize Zero123 [41] to predict four nearby views (10 degrees apart) of the input view. With these predicted views, we proceed to enumerate all possible elevation angles and compute the re-projection error for each candidate angle.

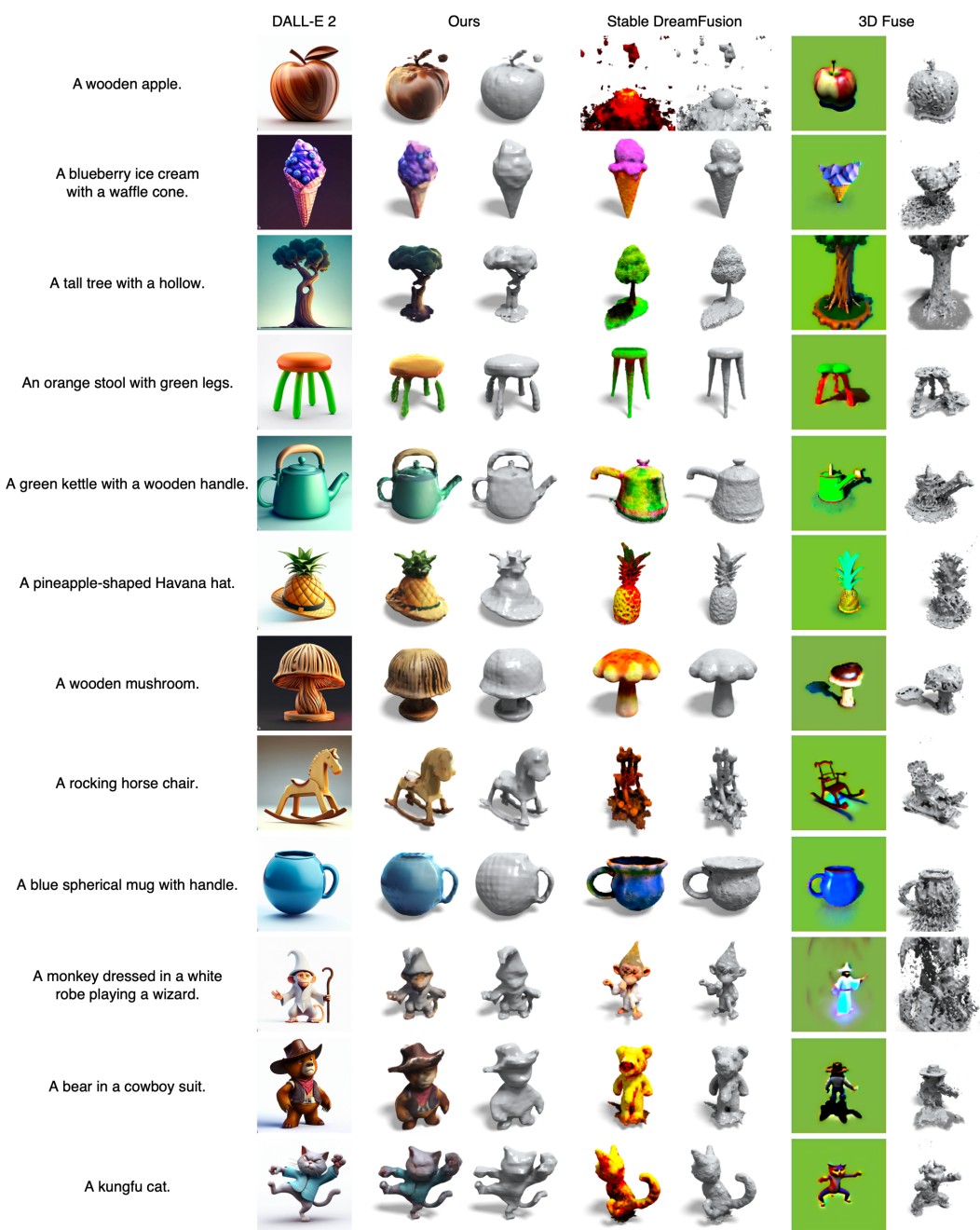

Figure 14: Text-to-3D: We compare our method against two native text-to-3D approaches Stable DreamFusion [60] and 3DFuse [72]. To enable text-to-3D, our method first uses a pretrained text-to-image model DALL-E 2 [63] to generate an image from input text (prompted with "3d model, long shot"), and then uplifts the image to a 3D textured mesh.

The re-projection error assesses the consistency between camera poses and image observations, akin to the bundle adjustment module employed in the Structure-from-Motion (SfM) pipeline.

Specifically, we enumerate all candidate elevation angles in a coarse-to-fine manner. In the coarse stage, we enumerate elevation angles with a 10-degree interval. Once we have determined the elevation angle $e^*$ associated with the smallest re-projection error, we proceed to the fine stage. In this stage, we enumerate elevation angle candidates ranging from $e^* - 10°$ to $e^* + 10°$ with a 1-degree

interval. This coarse-to-fine design facilitates rapid estimation, completing the elevation estimation module in under 1 second for each shape.

Given a set of four predicted nearby views, we perform feature matching to identify corresponding keypoints across each pair of images (a total of six pairs) using an off-the-shelf module LoFTR [74]. For each elevation angle candidate, we calculate the camera pose for the input image by employing the spherical coordinate system with a radius of 1.2 and an azimuth angle of 0. Note that the azimuth angle $\phi$ and the radius $r$ can be arbitrarily adjusted, resulting in the rotation and scaling of the reconstructed object accordingly. Subsequently, we obtain the camera poses for the four predicted views by incorporating the specified delta poses.

Once we have the four posed images, we compute the re-projection error by enumerating triplet images. For each triplet of images $(a, b, c)$ sharing a set of keypoints $P$, we consider each point $p \in P$. Utilizing images $a$ and $b$, we perform triangulation to determine the 3D location of $p$. We then project the 3D point onto the third image $c$ and calculate the reprojection error, which is defined as the $l1$ distance between the reprojected 2D pixel and the estimated keypoint in image $c$. By enumerating all image triplets and their corresponding shared keypoints, we obtain the mean projection error for each elevation angle candidate.

## E    Details of Training and Evaluation

**Training**    We train the reconstruction module using the following loss function:

$$\mathcal{L} = \mathcal{L}_{rgb} + \lambda_1 \mathcal{L}_{eikonal} + \lambda_2 \mathcal{L}_{sparsity} \tag{1}$$

where $\mathcal{L}_{rgb}$ represents the $l1$ loss between the rendered and ground truth color, weighted by the sum of accumulated weights; $\mathcal{L}_{eikonal}$ and $\mathcal{L}_{sparsity}$ are the Eikonal and sparsity terms, respectively, following SparseNeuS [45]. We empirically set the weights as $\lambda_0 = 1$, $\lambda_1 = 0.1$, and $\lambda_2 = 0.02$. For $\lambda_2$, we adopt a linear warm-up strategy following SparseNeuS [45]. To train our reconstruction module, we utilize the LVIS subset of the Objaverse [12] dataset, which consists of 46k 3D models across 1,156 categories. The reconstruction module is trained for 300k iterations using two A10 GPUs, with the training process lasting approximately 6 days. It is important to note that our reconstruction module does not heavily rely on large-scale training data, as it primarily leverages local correspondence to infer the geometry, which is relatively easier to learn and generalize.

**Evaluation**    We evaluate all baseline approaches using their official codebase. Since the approaches take only a single image as input, the predicted mesh may not have the same scale and transformation as the ground-truth mesh. To ensure a fair comparison, we employ the following process to align the predicted mesh with the ground-truth mesh. First, we align the up direction for the results generated by each approach. Next, for each generated mesh, we perform a linear search over scales and rotation angles along the up direction. After applying each pair of scale and z-rotation, we utilize the Iterative Closest Point (ICP) algorithm to align the transformed mesh to the ground-truth mesh. Finally, we select the mesh with the largest number of inliers as the final alignment. This alignment process helps us establish a consistent reference frame for evaluating the predicted meshes across different approaches. To calculate CLIP similarity, we render both ground-truth and generated meshes, capturing 24 views around the 3D shape from fixed viewpoints - 12 views at 30° elevation and 12 views at 0° elevation.

## F    Failure Cases and Limitations

Our method relies on Zero123 for generating multi-view images, which introduces challenges due to its occasional production of inconsistent results. In Figure 15, we present two typical cases that exemplify such inconsistencies. The first case involves an input view that lacks sufficient information, such as the back view of a fox. In this scenario, Zero123 struggles to generate consistent predictions for the invisible regions, such as the face of the fox. As a consequence, our method may encounter difficulties in accurately inferring the geometry for those regions. The second case involves an input view with ambiguous or complex structures, such as the pulp and peel of a banana. In such situations, Zero123's ability to accurately infer the underlying geometry becomes limited. As a result, our method may be affected by the inconsistent predictions generated by Zero123. It is important to acknowledge that these limitations arise from the occasional scenarios, and they can impact the

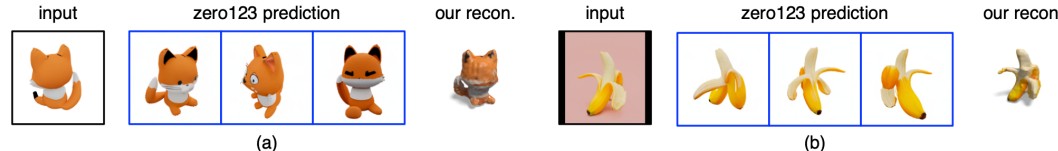

Figure 15: Failure cases. Our method relies on Zero123 to generate multi-view images, and we encounter challenges when Zero123 generates inconsistent results. (a) The input view lacks sufficient information. (b) The input view contains ambiguous or complicated structures.

performance of our method in certain cases. Addressing these challenges and refining the reliability of Zero123's predictions remain areas for further investigation and improvement.

We have also noticed slight artifacts on the back side of our generated results. As one of the first works in combining view-conditioned 2D diffusion models with generalizable multi-view reconstruction, we believe that there is still ample room for exploring more advanced reconstruction techniques and incorporating additional regularizations. By doing so, we expect to significantly mitigate the minor artifacts and further enhance results in the future.

