# OpenReview forum: "One-2-3-45: Any Single Image to 3D Mesh in 45 Seconds without Per-Shape Optimization"
_NeurIPS.cc/2023/Conference — NeurIPS 2023 poster_

### Official Review · Reviewer_eL63 · 2023-07-05

**Soundness:** 3 good
**Presentation:** 3 good
**Contribution:** 2 fair
**Rating:** 5
**Confidence:** 4

**Summary:**

This paper introduces a novel approach for converting a single image into a 3D textured mesh without the necessity for per-shape optimization. The method employs Zero-123 for generating novel views of the input image, which is then utilized by a generalizable MVSNeRF to reconstruct the 3D shape. To tackle issues like inconsistent multi-view predictions from Zero-123 and inaccurate camera poses, the authors have developed an elevation estimation module and specialized training strategies, including 2-Stage Source View Selection and Groundtruth-Prediction Mixed Training.

**Strengths:**

1. The paper proposes a method that reconstructs 3D shapes from a single image, eliminating the need for per-shape optimization. It does this through the effective application of feed-forward SparseNeuS.
2. The authors discuss and address several challenges associated with lifting Zero-123 predictions to 3D, such as inconsistent multi-views and inaccurate camera poses. The introduction of 2-Stage Source View Selection, Groundtruth-Prediction Mixed Training, and an Elevation Estimation Module appears to be effective solutions.
3. The results demonstrate that One-2345 enhances both the quality and efficiency of the single-image to 3D conversion. Additionally, the authors adeptly extend this to text-3D conversion using a pre-trained text-image model. The ablation studies provide strong evidence for the efficacy of the introduced modules and training strategies.

**Weaknesses:**

1. The paper mentions the use of SparseNeuS for reconstructing 3D shapes from multi-view predictions by predicting blending weights. The colors of the 3D points are computed as the weighted sum of projected colors. However, since SparseNeuS usually takes real images as input and this work uses potentially inaccurate multi-view predictions, I am concerned that directly aggregating colors from these predictions could lead to blurry textures and artifacts. I suggest the authors investigate the effects of aggregating or refining colors more thoroughly.
2. There is a noticeable discrepancy between training and inference. During training, the 2-Stage Source View Selection module generates multiple nearby views from ground truth anchor views for reconstruction. However, during inference, there are no ground truth anchor views available, and Zero-123 must rely on its own predicted anchor views, which may not always be accurate. The paper does not address these challenges or offer solutions. Additionally, the “2-Stage Source View Selection and Groundtruth-Prediction Mixed Training” section could benefit from clearer explanations.
3. SparseNeuS employs post-optimization after the feed-forward pass. It would be helpful to know if One-2345 adopts a similar strategy, and, if so, how much improvement can be attributed to such a lightweight post-optimization.
4. The overall concept of integrating a feed-forward MVSNeRF with Zero-123 is not groundbreaking. Moreover, the Groundtruth-Prediction Mixed Training, though practical, lacks novelty. However, the motivation for employing a feed-forward approach to elevate Zero-123 to 3D shape reconstruction is sound.

**Questions:**

see the strengths and weaknesses.

**Limitations:**

The paper does not discuss limitations.

---

> ### Author Rebuttal · Authors · 2023-08-10
>
> Thank you for your insightful comments and valuable suggestions. We will revise our paper based on your feedback. Here are our responses to your comments:
>
> **Weighted sum of projected colors**
>
> Thanks for pointing this out. We agree that computing the point colors as a weighted sum of projected colors may not be the optimal solution in our case, considering the underlying inconsistencies. We noted that some generalizable NeRF works (e.g., MVSNeRF) use an MLP to aggregate projected colors and directly predict point colors instead of their linear weights. We have launched an experiment to explore this strategy, and training is ongoing. We will update the results after the training is complete.
>
> **Discrepancy between training and inference**
>
> We also noticed this discrepancy. The high-level motivation is that each anchor image (one of eight) and its local images (four views, second stage) are expected to control a local region of the shape, and these eight regions should be relatively independent of each other. Since the local predictions for each anchor image are relatively accurate and the second-stage local predictions are also used in training, each local region should be reconstructed accurately. As a result, even if we utilize ground-truth anchor images during training, it should be possible to generalize to inconsistent predicted anchor images.
>
> In fact, we have experiments trying to utilize predicted anchor images during training. However, we found it challenging to supervise the training. For example, if we utilize ground-truth renderings to supervise predicted anchor images, it actually leads to worse results, as shown in Figure 8 of the main paper.
>
> We will clarify “2-Stage Source View Selection and Groundtruth-Prediction Mixed Training” in our revision, given more space.
>
> **Additional post-optimization**
>
> There are two ways to add post-optimization. The first one is fine-tuning with multi-view predicted images as done in the original SparseNeuS. This may help if the multi-view predictions are very consistent. However, we find that it typically does not improve our results due to underlying inconsistencies among multi-view predictions, as shown in Figure 2 of the PDF (see main rebuttal).
>
> Another strategy is to leverage priors from 2D diffusion models (e.g., StableDiffusion) to further fine-tune the generated mesh as in DreamFusion. Our generated results can serve as a good initialization and accelerate the optimization. While NeRF representation and volume rendering is used in the original DreamFusion to better handle topological changes, in our fine-tuning, mesh representation, and surface rendering can also be used to reduce the rendering time since there are no significant topological changes.
>
> **Integrating a feed-forward MVSNeRF with Zero-123**
>
> We would like to emphasize our contribution in proposing a new paradigm for generalizable single-image 3D generation, which overcomes the limitations of existing paradigms, such as efficiency and 3D consistency for optimization-based methods and poor generalization for 3D native generative models.
>
> It’s nontrivial to integrate a feed-forward MVSNeRF with Zero123. There still exists multiple challenges when combining them: (a) original SparseNeuS only focuses on frontal-view reconstruction while we need 360-degree full mesh reconstruction; (b) original SparseNeuS only considers consistent multi-view images as input instead of inconsistent multi-view predictions; (c) we need the camera poses of the input view for 3D reconstruction.
>
> We propose several critical training strategies and a pose estimation module to overcome these challenges. As highlighted by Reviewer L2hY, we demonstrate originality through our well-motivated and smart design choices and the underlying reasoning.
>
> **”Does not discuss limitations?”**
>
> We would like to clarify that we have discussed limitations in supplementary and will move them to the main text in our revision.

---

> > ### Author Response · Authors · 2023-08-17
> > **Updates on color prediction experiments**
> >
> > Thank you for your insightful suggestions. We have previously initiated an experiment in which we employ a MLP to aggregate projected colors, with the aim of directly predicting point colors rather than their linear weights. Recent experimental results indicate that this approach may help mitigate certain artifacts, such as white spots that can occur when exporting meshes. We agree that pursuing more sophisticated strategies for color prediction holds promise as a valuable avenue for future research.

---

> > ### Comment · Reviewer_eL63 · 2023-08-18
> >
> > The authors' rebuttal and additional experiments sufficiently address my main concerns. The post-optimization results in Figure 2 are interesting. I appreciate the authors taking the time to conduct these extra analyses. I am still interested to see results when replacing the blending weights with MLP prediction, as suggested in the rebuttal. This would provide further insight into the generalizability of One-2-3-4-5. The color blending scheme seems important for the strong generalizability shown in Figure 1. Seeing results with MLP-predicted colors could reveal if generalizability is dependent on the proposed blending approach or not. I will maintain my existing scores for now.

---

> > > ### Author Response · Authors · 2023-08-18
> > > **add links**
> > >
> > > Dear AC,
> > >
> > > Reviewer eL63 has requested additional figures from our experiments. The email guidelines suggest that including links might not be permissible. Could you please confirm if we are allowed to share an anonymous link?
> > >
> > > Thank you,

---

> > > > ### Author Response · Authors · 2023-08-20
> > > > **Can we add links?**
> > > >
> > > > Dear AC,
> > > >
> > > > Would you kindly confirm if we are permitted to share an anonymous link, in order to provide some figures to Reviewer eL63? I apologize for reaching out over the weekend, but the discussion phase ends on Monday at 1pm EDT.
> > > >
> > > > Thank you.

---

> > > ### Author Response · Authors · 2023-08-20
> > > **Response to Reviewer eL63**
> > >
> > > Dear Reviewer eL63,
> > >
> > > Thank you for your insightful feedback.
> > >
> > > In principle, the use of MLP-predicted colors should not compromise generalizability. Given that the projected pixel colors from all views serve as input to the MLP, it inherently has the capacity for internal "linear combination" or color blending. However, the MLP isn't strictly bound to this mechanism and might aggregate the color in more sophisticated ways.
> > >
> > > In our experiments, the inputs to the MLP comprise the coordinates of the query point, the corresponding interpolated cost volume feature, and the projected pixel colors from various views. As previously noted, employing the MLP approach yields similar results but addresses certain artifacts – like the white spots which can manifest during mesh exports.
> > >
> > > Our color-prediction MLP design is inspired by MVSNeRF, but we acknowledge there's potential for further refinement, especially by possibly integrating the view directions from all views as suggested in other related studies.
> > >
> > > We would like to provide visual results from our experiments. However, current guidelines, as per the email and the "FAQ for Authors" section, seem to restrict authors from sharing links within comments. We sincerely hope our textual explanation clarifies your queries. Should you have any further concerns or questions, please do not hesitate to let us know.

---

### Official Review · Reviewer_yirP · 2023-07-05

**Soundness:** 2 fair
**Presentation:** 2 fair
**Contribution:** 3 good
**Rating:** 4
**Confidence:** 4

**Summary:**

The authors tackle the problem of 3D reconstruction from a single image, which is challenging due to the lack of 3D information. They propose a novel method that combines a 2D diffusion model, Zero123, with a cost-volume-based 3D reconstruction technique, SparseNeuS, to generate a 360-degree 3D textured mesh in a feed-forward pass. They also estimate the elevation of the input shape and introduce several training strategies to improve the consistency and quality of the 3D mesh. Their contributions are:

* A novel method that leverages 2D prior models for 3D modeling without per-scene optimization.
* An elevation estimation module that computes the camera poses required by the reconstruction module.
* A series of essential training strategies that enable the reconstruction of 360-degree meshes from inherently inconsistent multi-view predictions.

**Strengths:**

* The work tackles 3D reconstruction from a single image, which is challenging and useful. The work can handle potentially any object category and generate a full 3D mesh from a single image.
* The proposed method achieves high-quality 3D reconstruction in a feed-forward manner without optimization, which is faster and more efficient. The work does not require per-scene optimization but relies on a single feed-forward pass to generate the 3D mesh and as such it is much faster than existing methods.
* The work leverages 2D prior models for 3D modeling, which is somewhat novel. It uses a 2D diffusion model to synthesize multi-view images of the input, and then uses multi-view 3D reconstruction techniques to obtain a 3D mesh.

**Weaknesses:**

* The evaluation section is noticeably light. The authors used 20 shapes from GSO and Objaverse to report F-Score and CLIP similarity. The number of shapes (20) is too low for any meaningful comparison considering that the method was trained with 46k 3D assets.

* The paper is framed that the work solves the image to 3D problem, which is true to some extend. In reality the work solve the problem of 3D lifting a single object, which is a much more constrained problem. I would advise the authors to change their wording to a more accurate description of the task.

* The work mentions that it used 46k Objaverse assets for training, i.e. the whole Objaverse-LVIS dataset . All figures have Objaverse-LVIS assets and they keep repeating in the figures (backpack, super mario, horse, minion etc). A quick search show nearly all of them, for example this is the backpack: https://skfb.ly/6XCoS . All other can be found with search on the website https://objaverse.allenai.org/explore . On the contrary, there a minimal results on assets not from objaverse which rises the question whether the reconstructions we are seeing in the paper are assets from the training set.

* The work has showcases a lot of licensed assets like pokemon, Super-mario and others. These assets require a licensing agreement with the respect companies that hold the rights. It might be sensible to replace them.

**Questions:**

* Would it be possible to do a human evaluation with hundreds of 3D assets comparing your results with that of Shap-E? This would enhance the quality of the current evaluation.

* Would it be possible to report numerical scores for hundreds of generated 3D assets rather than 20? To the same extend, can you report what the assets used were and whether they exist in Objaverse-LVIS?

* Objaverse created a licensing issue when it came out with many artists requesting to retract their 3D assets from the dataset. Did the authors respect this by checking the no-AI flag in Sketchfab for both training and evaluation?

+ Comments in Weaknesses

**Limitations:**

Yes to some extend

---

> ### Author Rebuttal · Authors · 2023-08-10
>
> Thank you for your insightful comments and valuable suggestions. We will revise our paper based on your feedback. Here are our responses to your comments:
>
> **Number of shapes for evaluation**
>
> We first would like to clarify that we used 40 shapes (20 from GSO and 20 from Objaverse, excluding the LVIS subset) instead of 20 shapes for evaluation. The limited number of evaluation shapes is mainly because many baseline algorithms are very time-consuming. For example, RealFusion takes ~90 minutes to generate a single shape on an A100 GPU. It's not practical to evaluate them on hundreds of shapes (e.g., it will take 2 A100 weeks for RealFusion to generate 200 shapes). Faced with these challenges, recent 3D AIGC papers have yet to quantitatively evaluate the method on large-scale datasets. For example, RealFusion evaluates the methods on only 21 shapes (7 categories) with one baseline, while we have 40 shapes and five baselines.
>
> **"Only reconstruct single objects, not single image to 3D?"**
>
> We agree that our current method mainly focuses on the generation of a single object. We have explicitly stated that in the abstract (line 6) and introduction (line 32). We will follow your suggestion to make the wording more rigorous.
>
> **”Results shown in the paper are from the training set?”**
>
> We would like to clarify that all results shown in the paper and supplementary are not seen during training of One-2-3-45, as stated in the supplementary. We split the Objaverse-LVIS dataset into training and validation sets and used only 43k of the 46k shapes for training. The backpack image is not from the LVIS subset. The images of super mario, the horse, and the minion are from somewhere other than the Objaverse dataset.
>
> **Report numerical scores for hundreds of generated 3D assets?**
>
> Since it's computationally expensive to evaluate optimization-based methods on hundreds of shapes, we add a comparison between the proposed method, Point-E, and Shap-E. We evaluate methods on 200 shapes from the GSO dataset, which doesn't contain Objaverse-LVIS shapes. The F-score for our method, Point-E, and Shap-E are 93.9, 91.5 and 93.3, respectively. The CLIP score for our method, Point-E, and Shap-E are 67.6, 65.1, and 68.7, respectively.
>
> **Objaverse license**
>
> Thanks for pointing this out. We utilize shapes with CC-BY 4.0 license for training and will add attribution in our paper.

---

> ### Comment · Area_Chair_RUdW · 2023-08-18
>
> Reviewer yirP,
>
> Please read the rebuttal provided by authors and raise a discussion if your concerns are not well addressed.
>
> Best,
> AC

---

### Official Review · Reviewer_AK29 · 2023-07-05

**Soundness:** 3 good
**Presentation:** 3 good
**Contribution:** 2 fair
**Rating:** 5
**Confidence:** 4

**Summary:**

The proposed method overcomes the challenges of lengthy optimization time, 3D inconsistency, and poor geometry that are common in existing methods. This method uses a view-conditioned 2D diffusion model, Zero123, to generate multi-view images from a single input image, and then lifts these images to 3D space. The 3D reconstruction is based on an SDF-based neural surface reconstruction method, with several training strategies proposed to enable the creation of 360-degree meshes. This method is faster, produces better geometry, and generates more 3D consistent results than existing methods. It has been evaluated on synthetic data and real-world images, demonstrating superior mesh quality and runtime. Additionally, it can be integrated with text-to-image diffusion models to support the text-to-3D task.

**Strengths:**

The main idea of this paper is to combine the view synthesis method Zero123 with the multi-view stereo (MVS) method SparseNeuS for 3D generation. This simple approach provides interesting insights and results, as it enables fast 3D generation that visually outperforms sophisticated 2D diffusion+NeRF methods like Dreamfusion, and also appears to be superior to other data-driven 3D generation methods such as Point-E and Shape-E.

The contribution of this paper is valid considering the advantages of runtime and better visual results.

**Weaknesses:**

- While simple and effective, the idea of the paper is not that eye-opening in the sense that it seems to be a natural extension or improvement of Zero-123.

- The performance of the proposed method is upper-bounded by the Zero-123. If Zeros-123 fails, it seems that there is no way for this method to generate reasonable output as well. In some sense, the problem of multi-view inconsistency is not solved but bypassed via the use of SparseNeuS.

- The area is advancing fast. Follow-ups of DreamFusion such as Magic3D and Fantasia3D have already achieved big improvements. It's not sure how the proposed method compares with more recent 2DDiffusion+NeRF methods.

**Questions:**

Going forward, which is a better path for 3D generation, novel view synthesis + mvs (like in this paper) or 3D data-driven (like in Shape-E)?

**Limitations:**

The issue of multi-view inconsistency is not addressed but bypassed. It would be nicer to have a method that is more fundamental to addressing this issue.

---

> ### Author Rebuttal · Authors · 2023-08-10
>
> Thank you for your insightful comments and valuable suggestions. We will revise our paper based on your feedback. Here are our responses to your comments:
>
> **Natural extension or improvement of Zero123?**
>
> We would like to emphasize our contribution in proposing a new paradigm for generalizable single-image 3D generation, which overcomes the limitations of existing paradigms, such as efficiency and 3D consistency for optimization-based methods and poor generalization for 3D native generative models.
>
> It’s nontrivial to extend Zero123 for 3D reconstruction in a feed-forward manner. Even in Zero123’s paper, they utilize optimization-based methods for 3D reconstruction. While it seems natural to combine Zero123 and generalizable NeRF methods, there still exists multiple challenges when combining them: (a) original SparseNeuS only focuses on frontal-view reconstruction while we need 360-degree full mesh reconstruction; (b) original SparseNeuS only considers  consistent multi-view images as input instead of inconsistent multi-view predictions; (c) we need the camera poses of the input view for 3D reconstruction.
>
> To overcome these challenges, we propose several critical training strategies and a pose estimation module. As highlighted by Reviewer L2hY, we demonstrate originality through our well-motivated and smart design choices and the underlying reasoning.
>
> **Multi-view inconsistency is not solved but bypassed.**
>
> We agree that multi-view inconsistency is still the main bottleneck and has not been fully addressed. However, our method is not tightly coupled with Zero123. Also, Zero123 is still the first few attempts in the direction of multi-view prediction. We believe that more progress will be made on 2D diffusion models for multi-view prediction, and any updates will also improve our method. Except when perfectly consistent multi-view prediction is possible, our proposed method is still effective and can be used to deal with small inconsistencies.
>
> **Comparison with more recent methods**
>
> We would like to point out that many 2D Diffusion+NeRF methods (including Magic3D and Fantasia3D) mainly support text conditions and cannot support the single image to 3D task natively. In the paper, we showed the result of Zero123+Stable DreamFusion, which differs from the original DreamFusion. Moreover, Magic3D has not released its code and uses an internal 2D diffusion model.
>
> We also would like to mention that we have compared the proposed method with most of the recent single image-to-text works: Point-E (Dec 22), RealFusion (March 23), 3DFuse (March 23), Zero123 (March 23), and even Shap-E (May 5).
>
> **Which is a better path for 3D generation**
>
> This is a great open question. It’s hard to say which path will dominate the task, as different applications may pose different requirements. We feel that multi-view prediction + 3D reconstruction (like in our paper) is a promising direction for open-world and time-sensitive applications. On the one hand, 3D native generative models (e.g., Shap-E) suffer from limited 3D data, while our method can benefit from richer 2D priors (e.g., millions of 3D shapes vs billions of 2D images), which makes open-world applications more possible. On the other hand, our method is much more efficient and better reserves 3D consistency than optimization-based methods (e.g., DreamFusion), which is important in time-sensitive applications (e.g., AR/VR, robotics, social media apps).
>
> It would be beneficial to combine various paths. For example, by using the generated results of our method as initialization, we can leverage optimization-based methods to fine-tune the results further and add more fine-grained details. With good initialization, optimization time should be significantly reduced, and 3D inconsistency issues should no longer exist. Furthermore, suppose we care about the internal structure of 3D shapes, it is better to combine our method with 3D native generative models since our method and optimization-based methods currently focus on the shape surface and cannot reconstruct the internal structure of a shape.

---

> > ### Comment · Reviewer_AK29 · 2023-08-11
> >
> > The authors give a fair response to my reviews. My score remains to be borderline accept.

---

> > > ### Author Response · Authors · 2023-08-18
> > > **Thank you**
> > >
> > > Thank you for your response and insightful reviews.

---

### Official Review · Reviewer_L2hY · 2023-07-05

**Soundness:** 3 good
**Presentation:** 3 good
**Contribution:** 3 good
**Rating:** 6
**Confidence:** 5

**Summary:**

This paper tackles the task of single image 3D reconstruction, where given a single input image of an object, the proposed method aims to generates a 360 degree 3D textured mesh.
Compared to prior works tackling a similar problem setting, this paper primarily aims to improve the quality of the reconstruction and reduce the time costed during inference time.

The proposed method contain 3 modules: (1) an off-the-shelf view-conditioned 2D diffusion model to generate multi-view images from a single input image, (2) a camera pose estimator, (3) an existing SDF-based multi-view reconstruction method to lift these images into 3D space.
The method is evaluated qualitatively and quantitatively on synthetic and real data, and outperforms most baselines.


**Strengths:**


Originality:

- This paper is well positioned in the literature. It properly summarizes and tackles existing limitations of related works with a similar task setting.
- This proposed method is a smart combination of existing methods, and the proposed design choices are well motivated.

Clarity:
- The writing is clear and easy to follow.

Quality:

- The qualitative and quantitative results outperform baselines and achieves SOTA. It still has a significant room for improvements. But it’s acceptable considering the challenges of the task.


Significance:

- The proposed method presents a novel idea for the literature -- incorporate a feed-forward model for diffusion-based methods.

**Weaknesses:**

- The novelty of the proposed method is relatively limited, as it heavily relies on prior works. Specifically, two out of the three modules of the proposed method (Zero123 and SparseNeuS) are built upon existing approaches. However, the authors do demonstrate originality through their design choices and the underlying reasoning for the combination of existing ideas.


- The proposed method contains several off-the-shelf or disconnected modules. It could have been beneficial to provide more discussion and analysis on how errors of one module affect the final performance.
For example, pose estimation is a task that is not completely solved, how will the pose estimation error affect the pipeline? For inference of real world images, how will the inaccuracy of segmentation masks affect the final quality?


- Although the quality of the final results achieved by the proposed method can be considered state-of-the-art, there is still significant room for improvement. From the qualitative results, most reconstructed 3D model is far from the quality of the input image. It can be helpful to expand the discussion of limitation to analyze why these artifacts can happen (e.g. blurriness, distorted shape).


**Questions:**

In general I find this paper above the acceptance bar. The questions below aim to elicit additional clarification and insights of the paper:

- If allowing an additional test-time optimization during inference time, will it further improve the quality of the outputs? (Or will it get harmed by the inconsistent multi-view images?)

- The authors mentioned the proposed method suffers less from the multi-face issue (the Janus problem). Is it a benefit of the prior work Zero123 or from original design of the proposed method?

- Minor:
L.98 generates -> generate
Description in L.150 “uses a spherical camera” and L.203 “a spherical camera model” is ambiguous.

**Limitations:**

The paper discusses failure cases in supp, including inconsistent multi-view images from Zero123, and artifacts on the back side. I encourage the authors to also explicitly mention in the main paper.

---

> ### Author Rebuttal · Authors · 2023-08-10
>
> Thank you for your insightful comments and valuable suggestions. We will revise our paper based on your feedback. Here are our responses to your comments:
>
> **How do errors in one module affect the final performance?**
>
> Correct segmentation mask and pose estimation are important for the final generation quality, as ablated in Figure 10.
>
> Provided that the multi-view prediction is accurate, our pose estimation module is inherently capable of estimating precise elevation angles. For example, if we are given four nearby ground truth views, the module should be able to estimate the pose accurately in most cases. Although the median error of the current estimation module is 5 degrees, we attribute this error primarily to the inconsistency of the multi-view prediction, which can be regarded as the internal error of Zero123 and can be improved by a stronger multi-view prediction module.
>
> The current segmentation models (e.g., SAM) are very powerful. They can generate very accurate segmentation masks with a small amount of human input. As a result, we don’t feel the segmentation module can be the bottleneck.
>
> Instead, the main bottleneck remains with Zero123 and the reconstruction module. While our method is somewhat tolerant of inconsistencies of multi-view predictions, our approach may still fail for the severe failure cases of Zero123. However, our method is not tightly coupled with Zero123. Also, Zero123 is still the first few attempts in the direction of multi-view prediction. We believe that more progress will be made on 2D diffusion models for multi-view prediction, which will also enhance our method.
>
> **Reasons for the artifacts**
>
> We have observed some systematic artifacts in the original results, such as waffle-like textures on the backside. We find that these can be fixed by replacing the depth loss with the mask loss during the training of the reconstruction module. We will update our results in the revision. For other types of artifacts (e.g., distortion and blurriness), we feel that the inconsistency of multi-view predictions mainly causes them since the results are much better when feeding with ground-truth multi-view renderings. We also improved the mesh rendering script used in the original submission so that the rendering can better reflect the colors of the generated mesh instead of dimming it out.
>
> **Additional test-time optimization**
>
> There are two ways to add test-time post-optimization. The first one is fine-tuning with multi-view predicted images as done in the original SparseNeus. However, we find that it typically does not improve the results due to underlying inconsistencies among multi-view predictions, as shown in Figure 2 of the PDF (see main rebuttal).
>
> Another strategy is to leverage priors from 2D diffusion models (e.g., StableDiffusion) to further fine-tune the generated mesh as in DreamFusion. Our generated results can serve as a good initialization and accelerate the optimization. While NeRF representation and volume rendering is used in the original DreamFusion to better handle topological changes, in our fine-tuning, mesh representation and surface rendering can also be used to reduce the rendering time since there are no significant topological changes.
>
> **Reasons for less multi-face issue**
>
> Yes, less multi-face issues mainly benefit from view-conditioned 2D diffusion (e.g., Zero123). Our reconstruction module also tolerates and fixes some small inconsistencies in Zero123's predictions.
>
> **Limitations**
>
> Thanks for your suggestion. We will move the limitation from the supplementary to the main text.

---

> > ### Comment · Reviewer_L2hY · 2023-08-16
> >
> > The authors rebuttal responded to my concerns. After reading the rebuttal and other reviews, I would keep my current rating of Weak Accept.
> >
> > The authors promised several parts for revision during the rebuttal phase – if accepted, please be sure to finish before the camera ready.
> > E.g.: “We improved our reconstruction module and fixed some notable artifacts on the backside. We will update our results in the revision.”; “We also improved the mesh rendering script used in the original submission”.

---

> > > ### Author Response · Authors · 2023-08-18
> > > **Thank you!**
> > >
> > > Thanks for your reply and insightful comment! We assure you that all the improvements and changes mentioned will be included in the revision.

---

### Official Review · Reviewer_Ph7H · 2023-07-07

**Soundness:** 3 good
**Presentation:** 2 fair
**Contribution:** 3 good
**Rating:** 6
**Confidence:** 5

**Summary:**

This paper introduces an efficient method for single-image 3D reconstruction that significantly improves upon previous techniques by generating textured mesh in one feed-forward pass. The authors employ a view-conditioned 2D diffusion model, Zero123, to create multi-view images from a single input view and then lift these images into 3D space using SparseNeuS. The proposed technique is much faster than existing methods, while reserving 3D consistency.

**Strengths:**

The proposed method enables the generation of 3D meshes in one feed-forward pass after training. Due to its training-based, one-pass approach, the generation speed significantly outperforms many concurrent optimization-based methods.

The quality of writing in the introduction and related work sections is commendable. The related work section covers the most recent work in the field and is well-organized.

**Weaknesses:**

The authors propose a multi-stage method primarily built upon the combination of multiple existing works. The work seems more engineering-oriented than technically innovative. Many aspects of the methodology, such as elevation estimation, 2-Stage Source View Selection, and Groundtruth-Prediction Mixed training, focus on bridging various components borrowed from existing baseline models.

Additionally, as the method relies on Zero123 and SparseNeuS, will it inevitably inherit their limitations? Can this method circumvent the failure cases of Zero123 or SparseNeuS? The method 12345 is trained on 1,156 categories from the Objaverse-LVIS dataset. How does 12345 perform when the images fall outside these 1,156 categories? Many concurrent image-to-3D works can generalize to uncommon objects due to their large diffusion models. Could 12345's training in aligning Zero123 and SparseNeuS diminish the zero-shot capability?

The authors provide limited visualization cases in their ablation study. Including quantitative results to support the ablation study would be beneficial, as 3D generation is typically unstable and hard to reproduce.

The organization and presentation of the methodology section could be improved. The method involves multiple stages, but the authors do not strictly adhere to the method structure order and insert discussions throughout. This makes sections 3.1 and 3.3 difficult to follow.

From the visualization, the mesh generation quality is not so impressive.

Could the authors explain their motivation for learning SDF and RGB from two separate MLPs? Many NeRF papers typically predict these from the same network.

**Questions:**

See the Weaknesses above.

**Limitations:**

Despite the concerns raised in the weaknesses section, a positive review score is given in recognition of the effort to formulate a single-pass pipeline and for the considerable credit on the superior generation efficiency.

---

> ### Author Rebuttal · Authors · 2023-08-10
>
> Thank you for your insightful comments and valuable suggestions. We will revise our paper based on your feedback. Here are our responses to your comments:
>
> **Perform outside 1,156 categories?**
>
> Yes, our method can generate meshes for images  beyond the scope of the 1,156 training categories, as shown in Figure 1 of the attached PDF (see main rebuttal). Our 3D reconstruction module inherently leverages local correspondences for reconstruction and is thus designed to have a robust generalization ability to unseen data. This characteristic ensures that our method retains its zero-shot capability.
>
> **Circumvent the failure cases of Zero123 or SparseNeuS?**
>
> Our method demonstrates a certain level of tolerance towards the inconsistencies present in Zero123's multi-view predictions. In contrast, traditional optimization-based NeRF methods fail completely, as shown in Figure 3.
>
> We agree that our approach may also fail for the severe failure cases of Zero123. However, our method is not tightly coupled with Zero123. We believe that more progress will be made in 2D diffusion models for multi-view prediction,  which will enhance our approach.
>
> **Bridging various components**
>
> It’s nontrivial to combine 2D diffusion models and 3D reconstruction, and there exist many challenges. As highlighted by Reviewer L2hY, we demonstrate originality through our well-motivated and smart design choices and the underlying reasoning.
>
> Also, we would like to emphasize our pivotal contribution in proposing a new paradigm for generalizable single-image 3D generation, which effectively addresses the limitations inherent in existing paradigms, such as efficiency and 3D consistency issues for optimization-based methods and poor generalization for 3D native generative models.
>
> **Limited examples in ablation studies**
>
> Thanks for the suggestion. We will add more qualitative examples and quantitative results in our revised version of supplementary materials.
>
> **Why two separate MLPs?**
>
> Many NeRF papers use two separate MLPs [1,2,3] because the SDF/density branch only takes XYZ coordinates as input, while the RGB branch may include both XYZ coordinates and view directions. It is common for them to share some feature networks (same as ours), but the final decoders are usually different.
>
>
>
> **Generation quality is not so impressive?**
>
> We improved our reconstruction module and fixed some notable artifacts on the backside. We will update our results in the revision.
>
>
>
>
> [1] Chen, Anpei, et al. "Tensorf: Tensorial radiance fields." European Conference on Computer Vision. Cham: Springer Nature Switzerland, 2022.
>
> [2] Mildenhall, Ben, et al. "Nerf: Representing scenes as neural radiance fields for view synthesis." Communications of the ACM 65.1 (2021): 99-106.
>
> [3] Wang, Peng, et al. "Neus: Learning neural implicit surfaces by volume rendering for multi-view reconstruction." arXiv preprint arXiv:2106.10689 (2021).

---

> > ### Comment · Reviewer_Ph7H · 2023-08-19
> > **Response to the rebuttal**
> >
> > Thanks to the authors for the detailed rebuttal. I have no further questions at this point, and I hope the authors can improve their future version as promised. I have raised my score to weak accept.

---

### Author Rebuttal · Authors · 2023-08-10

Dear Reviewers,

Thank you for dedicating your time to review our paper and offering insightful feedback. We sincerely appreciate your efforts to help enhance the quality of our research. We are also pleased to note that all reviewers were supportive of our work:

(a) Recognize our contribution in proposing a novel single-pass paradigm for 3D generation (Ph7H, L2hY, yirP, eL63), which completely differs from existing 3D AIGC paradigms of optimization-based methods and 3D native generative models.

(b) Praise our paper is well-positioned in the literature and properly summarizes and tackles existing limitations of related works (Ph7H, L2hY, AK29), including lengthy optimization time, 3D inconsistency, poor geometry, and poor input adherence.

(c) Acknowledge our superior generation efficiency (Ph7H, AK29, yirP, eL63) and outperforming previous techniques (Ph7H, L2hY, AK29, eL63).

(d) Acknowledge our efforts in addressing several challenges associated with lifting Zero123 predictions to 3D, and find our solutions and design choices to be effective (eL63), smart, and well-motivated (L2hY).

(e) Praise our insightful ablation study, discussion, and underlying reasoning (L2hY, AK29, eL63).

(f) Find our paper well-written and easy to follow (L2hY, Ph7H for introduction and related work).

---

### Decision · Program_Chairs · 2023-09-21

**Decision:**

Accept (poster)

**Comment:**

The paper received positive feedbacks. Despite the novelty is a major concern as it is basically a combination of prior works, the quality and speed are superior to prior works. Most of the concerns are addressed in the rebuttal. Please revise the final manuscript accordingly as promised.